# SPING Block Analgesia in Non-Operative Management of Proximal Femur Fractures in Older Adults Living with Frailty: A Retrospective Cohort Study

**DOI:** 10.3390/geriatrics10010010

**Published:** 2025-01-10

**Authors:** Miriam W. A. van der Velden, Thamar Kroes, Nick J. G. Visschers, Frank P. J. F. de Loos, Pleun Janssens, Bart Spaetgens, Miriam C. Faes, Marieke H. J. van den Beuken-van Everdingen, Arnela Suman

**Affiliations:** 1Department of Pain Medicine and Palliative Care, Amphia Hospital, 4818 CK Breda, The Netherlandsnvisschers@amphia.nl (N.J.G.V.);; 2Department of Trauma Surgery, St. Antonius Hospital, 3435 CM Utrecht, The Netherlands; 3Department of Orthopedic Surgery, Amphia Hospital, 4818 CK Breda, The Netherlands; 4Geriatrics Section, Department of Internal Medicine, Maastricht University Medical Center, 6229 HX Maastricht, The Netherlands; 5Department of Geriatrics, Amphia Hospital, 4818 CK Breda, The Netherlands; 6Palliative Care Section, Department of Internal Medicine, Maastricht University Medical Center, 6229 HX Maastricht, The Netherlands

**Keywords:** palliative care, geriatrics, hip fracture, pain management, spinal analgesia, anesthesia

## Abstract

**Background/Objectives**: Spinal Phenol IN Glycerol (SPING) block is a novel palliative pain treatment for the non-operative management of proximal femur fractures (PFFs) in older adults living with frailty. Effective pain management that aligns with patient preferences and minimizes opioid use is critical in this setting. This study evaluated the patient, safety, and process outcomes of SPING block in this population. **Methods**: A retrospective cohort study was conducted in a suburban teaching hospital from March 2021 to June 2024, which included sixty-eight older adults living with frailty that suffered from a PFF and received SPING block. Data were collected from the Electronic Patient Records. The patient living situation was visualized with a Sankey diagram. Changes in pain scores and opioid use were assessed using the Wilcoxon Signed Rank test. **Results**: The median patient age was 89 years (Interquartile range (IQR) 83–92). Most were severely or terminally ill (American Society of Anesthesiologists (ASA) ≥ 4, 72%) and had cognitive impairment or dementia (68%). SPING block was effective in 93% of patients, significantly reducing median pain scores (4 [IQR 3–5] to 0 [IQR 0–1], *p* < 0.001) and opioid use (15 mg/day [IQR 4–30] to 0 mg/day [IQR 0–0], *p* < 0.001). Within 24 h, 84% could sit upright and 44% could transfer between their bed and chair. The median time to discharge was one day (IQR 0–3), with a median survival of 13 days (IQR 7–44). **Conclusions**: This study supports SPING block as a viable option for older adults living with frailty suffering from a PFF who opt for non-operative management in a palliative setting. SPING block for PFFs in a palliative setting offers effective pain relief, reduces opioid use, and enables mobility for older adults living with frailty. Follow-up is essential to monitor efficacy and safety. Prospective studies are needed to confirm these findings.

## 1. Introduction

The incidence rate of proximal femur fractures (PFFs) is expected to increase with the increase in the aging population. Proximal femur fractures involve the proximal end segment of the femur. While surgical interventions remain critical for PFF treatment, it can be high risk in older adults living with frailty and may not contribute to quality of life [1,2,3,4,5]. Recent evidence suggests that non-operative management (NOM), after shared decision-making (SDM), is an ethical and viable treatment alternative for older adults living with frailty with a limited life expectancy [6]. SDM is a collaborative process where patients and clinicians make healthcare decisions together, incorporating evidence-based information and patient preferences. However, the NOM of a PFF requires relatively high doses of opioids, which can lead to adverse effects such as delirium and decreased consciousness [7,8]. These side effects can reduce the quality of life during the palliative phase, highlighting the urgent need for improved pain management in older adults living with frailty suffering from a PFF and receiving NOM [7,8,9]. Developing pain management strategies that align with patient preferences, minimize surgical reliance, and decrease prolonged opioid use is crucial [10,11].

Regional techniques, such as femoral nerve blocks or femoral catheters, have been shown to decrease pain intensity and opioid use while improving functional outcomes [12,13,14,15,16,17,18]. However, the effectiveness of these techniques in the long term remains uncertain. Regional blocks are primarily intended for temporary pain relief, while catheters designed for more prolonged pain management often fail or are unavailable outside hospital settings. In the palliative phase, the primary treatment goal is improving the quality of life rather than functional recovery. For these patients, more comprehensive pain management that does not prioritize preserving motor function may be a treatment option [19].

In response, we introduced the Spinal Phenol IN Glycerol (SPING) block, a one-time and definitive intervention, which involves a single-shot of intrathecal phenol–glycerol [19,20,21]. It has shown promising results in older adults living with frailty suffering from a PFF and receiving NOM, providing immediate pain relief and rapid hospital discharge [20]. No robust data are available regarding SPING block application in palliative care; therefore, an observational retrospective cohort study was conducted to assess pain intensity, mobility, and opioid consumption. Secondly, we assessed the safety, process, and long-term patient outcomes.

## 2. Materials and Methods

### 2.1. Study Design and Setting

This observational, retrospective cohort study was conducted at a suburban teaching hospital, spanning from March 2021 to June 2024. The study did not include a control group due to the observational, retrospective design of the study as well as the SPING block being standard care in the study hospital. It was not deemed ethical to deny patients local pain management for study purposes. All patients aged 70 years or older with a radiologically confirmed PFF who were treated with the SPING block were included in this study.

### 2.2. Procedures

All patients were treated according to standard hospital protocol. Upon admission to the emergency department (ED), patients were initially assessed by the attending surgical resident. A geriatrician performed a comprehensive geriatric assessment and judged whether a patient was frail. Frailty was defined as a condition characterized by living in nursing home environments or receiving equivalent care, in combination with severe comorbidities, mobility problems, or malnutrition. If the patient was identified as frail and there was uncertainty about the appropriateness of surgical treatment among the patient, their representatives, or the physicians, an SDM process was initiated within 24 h. This process involved evaluating the benefits and risks of surgical treatment, and the likelihood of regaining meaningful mobility against the patient’s individual needs, values, and perioperative risks using the ASA score. If NOM was preferred after the SDM, pain management options were discussed with the anesthesiologist/pain specialist, including oral opioids according to standard hospital protocols and SPING block. If an interest in SPING block was expressed, the benefits and risks, i.e., loss of mobility in the affected extremity, of SPING block were discussed in an SDM process. Informed consent for treatment was obtained in all cases from the patient or their representative in the cases where the patient was judged to be mentally incompetent to decide on the treatment of the PFF.

### 2.3. Treatment Protocol

After informed consent for treatment was obtained, SPING block was administered in the recovery room, ideally within 24 h of admission to the ER. The procedure consisted of a single shot of spinal anesthesia using 0.8 mL of a 10% phenol–glycerol mixture, administered with the patient in a lateral position. The SPING block effectiveness was assessed within 30 min after treatment and after the sedation effects subsided. An effective SPING block was defined as the absence of pain when the affected leg was flexed at 90 degrees, as determined through direct patient feedback by asking for the pain score or, when direct feedback was not possible, by asking the patient about experienced pain intensity while observing facial expression and body movements. Slow-release opioids were discontinued immediately after the SPING block. Patients were discharged at the earliest appropriate time, depending on their clinical condition.

### 2.4. Data Collection and Outcomes

The data for this study were collected retrospectively from the Electronic Patient Records (EPRs). The dataset included clinical, safety, and process outcomes. The primary outcomes of this study were pain intensity, opioid consumption, and mobility. The secondary outcomes were safety outcomes (i.e., effectiveness, repeat procedures, and complications), process outcomes (i.e., admission duration and discharge location) and long-term patient outcomes (i.e., survival and follow-up outcomes). Details regarding the operationalization of the data, the measurement instruments used, and the timing of data collection are outlined in Table 1. Detailed information on all the study variables is included in Appendix A.

### 2.5. Follow-Up

A follow-up telephone call was made six weeks after the treatment. The patients or their representatives were asked for any residual pain, their overall satisfaction with the SPING block treatment, complications, and, if applicable, the date of death to assess mortality. In the cases where the patient was cognitively impaired or were deceased, representatives provided the information. In all cases where no routine six-week follow-up consultation was conducted, the patient or their representative was contacted by phone during the data collection of this study to complete the follow-up.

### 2.6. Statistical Analysis

Descriptive statistics were used to characterize the study population. All continuous variables were assessed for normality using histograms and the Shapiro–Wilk test (α < 0.05), which confirmed a non-parametric distribution. Accordingly, the results are reported as the median with the interquartile range (IQR). The Wilcoxon Signed Rank test (confidence level: 95%; α < 0.05) was used to evaluate the changes in pain scores and opioid use before and after the SPING block treatment. The statistical analysis was performed using IBM SPSS Statistics version 29.0.0.0. The Sankey diagram was created using sankeymatic.com [24].

### 2.7. Ethics

This research report followed the Strengthening the Reporting of Observational Studies in Epidemiology (STROBE) Guidelines for reporting observational studies [25]. The institutional review board of Amphia Hospital approved the study with a waiver of consent (N2021-0482).

## 3. Results

### 3.1. Patient Characteristics

A total of 68 patients were included in the study, with a median age of 89 years (IQR 83–92) and 63% (*n* = 43) of the patients were female. Cognitive impairment was prevalent in 68% (*n* = 46) of the patients, with 9% (*n* = 6) having mild cognitive impairment and 59% (*n* = 40) being diagnosed with dementia.

A majority of the patients (85%; *n* = 58) was dependent on care for activities of daily living (ADL). Additionally, 88% (*n* = 60) of the patients had a pre-fracture mobility score of ≤2 on the Functional Ambulation Category (FAC) scale. Approximately 72% (*n* = 49) of the patients were classified as ASA (American Society of Anesthesiologists) IV and 28% (*n* = 19) were classified as ASA III. The most prevalent fracture location was the femoral neck, accounting for 56% (*n* = 38) of the cases. Pertrochanteric fractures constituted 28% (*n* = 19) of the cases, periprosthetic fractures accounted for 15% (*n* = 10), and subtrochanteric fractures accounted for 2% (*n* = 1). The patient characteristics are provided in Table 2.

### 3.2. Pain Intensity and Opioid Consumption

The median Numeric Rating Scale (NRS) pain score before the SPING block was 4 (IQR 3–5), which decreased significantly to 0 (IQR 0–1) immediately after the treatment (*p* < 0.001). The average NRS over the first 24 h after the SPING block was 0.4 (median; IQR: 0–1) and the median NRS at discharge was 0 (IQR 0–1). The median Morphine Milligram Equivalent consumption decreased significantly from 15 mg per day (IQR 4–30) before treatment to 0 mg per day (IQR 0–0) post-treatment (*p* < 0.001). All the SPING block outcomes are reported in Table 3.

### 3.3. Mobility Outcomes

The majority of the patients (84%; *n* = 57 were able to sit upright within 24 h post-treatment. The observed mobility at discharge was walking a couple of steps in 3% (*n* = 2) of the patients, followed by ability to perform bed–chair transfers was observed in 44% (*n* = 30) of the patients, sitting upright in 22% (*n* = 15) of the patients, and 31% (*n* = 22) of the patients remained bedridden.

### 3.4. Safety Outcomes

The SPING block proved effective after the first treatment in 93% (*n* = 63) of the patients; three of these patients required a second dose within the same treatment session. A second SPING block procedure was performed in five patients, which subsequently resulted in adequate analgesia. The time between the first and second SPING procedures was not available in this study.

The reported complications during hospital admission after the SPING block included hypotension in 6% (*n* = 4), fecal incontinence in 4% (*n* = 3), urinary incontinence in 3% (*n* = 2), and fever in 3% (*n* = 2) of the patients. One patient experienced cardiac decompensation one day after the SPING block. Anemia or bleeding complications were not reported.

### 3.5. Process Outcomes

The SPING block was administered within a median of one day (IQR 0–2) after admission to the ED. The median time from the SPING block until discharge was one day (IQR 0–3). Approximately 63% (*n* = 43) of the patients were discharged to their pre-fracture living environment, while 7% (*n* = 5) died in hospital. Among the 20 remaining patients, appropriate care was not available in their pre-fracture living situation: 9% (*n* = 6) of the patients were discharged to a hospice, 15% (*n* = 10) of the patients required additional care in a nursing home, and 6% of the patients (*n* = 4) were discharged to geriatric rehabilitation. Figure 1 provides an overview of the patients’ living situation before and after the SPING block.

**Figure 1 geriatrics-10-00010-f001:**
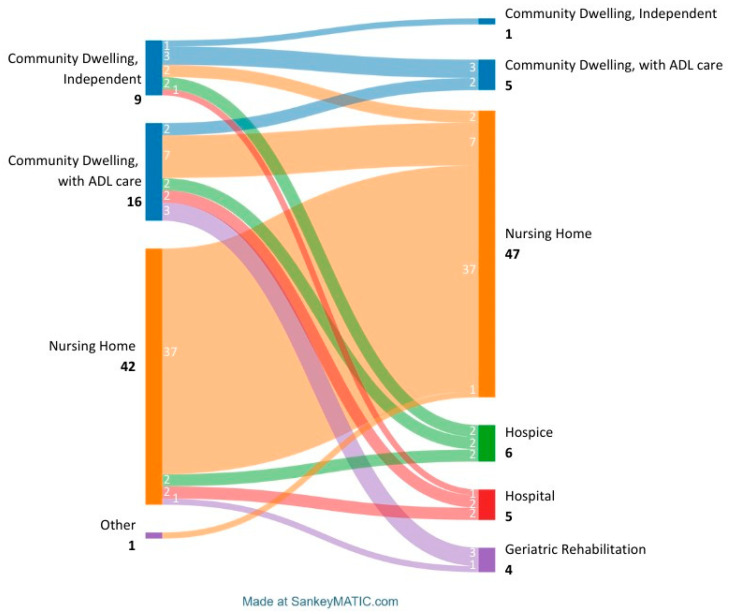
N patients per living situation before and after proximal femur fracture.

### 3.6. Long-Term Patient Outcomes

The median survival after the SPING block was 13 days (IQR 7–44), with a 30-day mortality rate of 47% (*n* = 32) and a 90-day mortality rate of 59% (*n* = 40).

Follow-up was conducted at a median of 146 days (IQR 40–372) after the SPING block. In 22% (*n* = 15) of the patients, residual pain in the affected hip was reported. This residual pain was described as mild and not requiring additional opioid consumption, except in two cases where opioids had remained indispensable for patient comfort.

The complications reported at follow-up included one patient experiencing knee pain in the affected leg, and one case of slight unilateral groin pain. Two patients reported neuropathic pain; in one patient, this may have been due to the reduced efficacy of the SPING block, and the other patient reported neuropathic pain 56 days after the treatment.

Overall, 81% (*n* = 55) of the patients or their representatives were satisfied with the SPING block treatment. Dissatisfaction with SPING block was rooted in insufficient pain relief directly after discharge. This was attributed to a too rapid discharge after the treatment, resulting in increased pain intensity at home.

## 4. Discussion

The findings of this study indicate that SPING block is able to provide substantial pain relief for older adults living with frailty suffering from a PFF and receiving NOM. The significant reduction in both pain scores and opioid consumption observed in this cohort underscores the potential of SPING block as a viable alternative to conventional pain management strategies that rely on opioids. Furthermore, SPING block was sufficient after the first treatment in the majority of cases, with no complications observed that impaired the patients’ quality of life. These results are particularly relevant in the context of palliative care for patients receiving NOM, where the quality of life and comfort are paramount across various fracture types.

SPING block appears able to provide immediate and prolonged pain relief, accompanied with a substantial decrease in opioid consumption. The pain reduction was highly clinically relevant with patients achieving a median NRS of 0 (IQR 0–1), and maintaining a low NRS of 0.4 (IQR 0–1) over the first 24 h. This indicates minimal to no pain, potentially enhancing the patients’ quality of life. The associated decrease in MME mitigates critical concerns related to opioid-induced adverse effects, such as delirium and decreased consciousness [9,15]. A decreased opioid need may improve quality of life through increased consciousness, which facilitates contact with loved ones. Importantly, the SPING block enabled mobility, as evidenced by the ability of patients to sit upright within 24 h post-treatment. Not all patients who were physically able to sit upright did so at discharge. This may be due to side-effects or death during the hospital stay.

For this study, all clinical events after the SPING block were transparently documented as potential complications that could be associated with the procedure. The safety evaluation of the SPING block identified possible complications, including urinary and fecal incontinence, hypotension, fever, and neuropathic pain. The patients or their representatives were not routinely questioned about specific complications and the patients were not routinely physically examined in the palliative care setting. Consequently, only clinically significant events were likely reported, potentially leading to an underestimation of other (likely minor) complications. However, complications that significantly affected quality of life were likely identified during the follow-up. The retrospective study design limited us in the analysis of potential complications and their direct link to the SPING block. It should also be noted that many patients in the target population may have already been experiencing certain health challenges, such as urinary or fecal incontinence, before the SPING block. Despite these limitations, our findings suggest that SPING block may be a worthwhile and safe approach for managing acute pain in older adults living with frailty suffering from a PFF who have opted for NOM.

SPING block appears to be efficient and feasible for older adults living with frailty that are suffering from a PFF. This was demonstrated by the short interval from hospital admission to the administration of the SPING block, followed by rapid discharge. Most patients were able to transition to a nursing home or hospice care within a day of treatment, underscoring the SPING block’s role in facilitating appropriate and timely care transitions. Although a short hospital stay is a notable feature of SPING block in palliative care, we recommend careful consideration prior to discharge and follow-up pain assessments. It is crucial to prevent unexpected pain complaints after discharge, which could necessitate additional medical interventions or even hospital readmission.

SPING block is a one-time, definitive intervention. As such, no ongoing management is required after the procedure. This represents one of the key advantages of SPING block compared to other nerve blocks or catheter-based approaches that require continuous pain medication. These alternatives often face challenges such as catheter dislocation or obstruction, which can necessitate the return of a highly frail patient population—often from their home or nursing home—to the hospital for further intervention.

Satisfaction with the SPING block was high, which may indicate that it is an appropriate treatment in NOM of various fracture types in older patients living with frailty. SDM is important to ensure that SPING block aligns with the patient care goals [26,27]. The observed mortality rates, while consistent with the frailty and health status of the cohort, emphasize the palliative nature of NOM of a PFF [6,28,29]. Almost half of the patients were deceased within 30 days, and the survival times varied among those who lived longer. This variability underscores the uncertainty in outcomes for patients with limited life expectancy treated with SPING block, with no treatment dissatisfaction reported regarding prolonged life duration. All patients were discharged to facilities that matched their anticipated care needs, although not all patients returned to their pre-fracture living environments. This underlines the importance of timely discussions about patient preferences and treatment goals, ensuring that treatment plans are well-aligned with available and viable treatment options, thereby enhancing the quality of care without resorting to unnecessary and invasive treatments [30,31,32,33].

Treatment modalities for NOM for older adults living with frailty suffering from a PFF continue to evolve through the development of definitive Regional Nerve Blocks. For instance, the Pericapsular Nerve Group (PENG) block and Posterior Hip Pericapsular Neurolysis (PHPN) have been described for this purpose [18,34,35,36,37]. Both SPING block and Regional Nerve Blocks significantly alleviate pain immediately post-treatment and reduce opioid usage [20,38]. Nevertheless, SPING block may offer more sustained and extensive pain relief in specific situations due to its broader impact on neural pathways [39]. SPING block might be a more appropriate treatment for patients who are already bedridden or for those with fracture types that are more distal from the hip capsule [40].

This study has several strengths and limitations. The key strengths include the uniform application of SPING block by specialized anesthesiologists. Building on an initial case series of 10 patients [20], this study is the first of its scale, with 68 patients, allowing for preliminary evaluations of the efficacy and safety of SPING block. The study was conducted among the target population of the intervention, ensuring generalizability in this small population. To improve consistency in the retrospective evaluation, the pain scores were averaged over 24 h.

The study has several limitations due to its single-center, retrospective, and observational design. As patient selection was based on SDM, the study population was heterogenous; this may have introduced an unknown bias into the assessment of the SPING block efficacy. As SPING block is a standard of care in the study hospital, a control group was not feasible in this study. It was not feasible to distinguish pain scores during movement versus rest retrospectively, which may have compromised the interpretation of the SPING block efficacy. The retrospective design also hindered complication monitoring and accurate reporting of pre-existing conditions, such as incontinence. Furthermore, it was necessary to calculate certain endpoints (using FAC) based on known retrospective data the retrospective assessments of comorbidity could have resulted in underreporting. However, this likely occurred randomly, and it is well-known that comorbidity has been recognized as valid when assessed retrospectively [41,42]. The rapid discharge resulted in the absence of comprehensive data on long term care needs, pain control, and complications. Mortality data were not available for all patients, which may have resulted in an underestimation of mortality.

It is important to note that many patients included in the study had cognitive impairment or dementia, conditions known to impair pain reporting [43,44]. Consequently, there may have been an underreporting of pain and an overestimation of the effectiveness of the SPING block in this study. However, older adults living with frailty are likely to be in need of non-operative interventions like SPING block due to the higher surgical risks that come with frailty and were therefore included in the study [45].

Larger prospective studies are essential to refine the efficacy of SPING block and its implications, such as increased consciousness, for quality of life. Transparent patient selection for NOM remains important. Future research should ensure accurate pain measurements in patients with cognitive impairment or dementia, detailed mobility observation, and close monitoring of potential complications. Additionally, research should also explore the safety of SPING block and variability in treatment effectiveness, and compare different pain management strategies in NOM of a PFF. Patient follow-up in a study context should be conducted with attention to quality of life. These studies are necessary to enhance our understanding of optimizing care and quality of life across different settings and tailoring care to individual needs in older adults living with frailty suffering from a PFF.

## 5. Conclusions

SPING block may be a viable option for older adults living with frailty suffering from a PFF that are opting for NOM in a palliative setting. Patients with various fracture types showed significant pain reduction with a decreased opioid need after SPING block. SPING block may support quality of life by decreasing opioid needs and enabling patients to reach a sitting or transfer position. These results are particularly relevant in the context of palliative care, where quality of life and comfort are paramount. Adequate follow-up is necessary after SPING block to monitor the treatment efficacy and potential complications. Prospective research is warranted to confirm and refine the results of this retrospective observational cohort study.

## Figures and Tables

**Table 1 geriatrics-10-00010-t001:** Outcome measures, operationalization, measurement instruments, and time points.

	**Measurement Instrument and Operationalization**	**Measurement Time Points**
**Pain intensity**	Numeric Rating Scale (NRS) [22]. All measurements available from Electronic Patient Records	(1) Average in 0 to 24 h before SPING block treatment; (2) immediately after SPING block treatment; (3) average in 0 to 24 h after SPING block; (4) at discharge
**Opioid consumption**	Morphine Milligram Equivalents (MME) [23]	(1) Sum of MME in 0 to 24 h before SPING block treatment; (2) sum of MME in 0 to 24 h after SPING block treatment
**Ability to sit upright**	(Yes or No)	Tested directly after SPING block treatment
**Mobility at discharge**	Bedridden; sitting upright; bed-chair transfers; walking a few steps	Most advanced observed mobility during admission after SPING block
**SPING block effectiveness**	Paresis, no light touch sensation, and passive extension/flexion without pain response in the affected upper leg (Yes or No)	Within 30 min after SPING block treatment and after sedation has worn off
**Complications**	Urine incontinence, fecal incontinence, hypotension, fever, paresis of other leg, neuropathic pain, or other complications	(1) During admission; (2) at follow-up

**Table 2 geriatrics-10-00010-t002:** Patient characteristics.

**Characteristic**	**Value**	**Missing**
Age, median (IQR), y	89 (83–92)	0
Female	43 (63.2)	0
Cognitive impairment		1 (1.5)
None	21 (30.9)	
Mild cognitive impairment	6 (8.8)	
Dementia	40 (58.8)	
Pre-fracture living situation		0
Community dwelling, independent	9 (13.2)	
Community dwelling, with ADL care	16 (23.5)	
Nursing home	42 (61.8)	
Other	1 (1.5)	
FAC		1 (1.5)
0	10 (14.7)	
1	26 (38.2)	
2	24 (35.3)	
3	2 (2.9)	
4	5 (7.4)	
ASA Classification		0
3	19 (27.9)	
4	49 (72.1)	
Fracture characteristics		0
Femoral neck location	38 (55.9)	
Pertrochanteric	19 (27.9)	
Subtrochanteric	1 (1.5)	
Periprosthetic	10 (14.7)	1
Vancouver A	2 (20)	
Vancouver B	7 (70)	
Vancouver C	0	

Note: Data are given as number (percentage), unless otherwise indicated. Abbreviations: IQR, Inter Quartile Range (denoted as p25–p75); FAC, Functional Ambulation Category; ASA, American Society of Anesthesiologists; SPING, Spinal Phenol IN Glycerol.

**Table 3 geriatrics-10-00010-t003:** Patient outcomes before and after SPING block.

**Outcome**	**Value**	**Missing**
Pain Intensity		
Median NRS (IQR) before SPING block	4 (3–5)	3 (4.4)
Median NRS (IQR) after SPING block		
Immediate	0 (0–1)	3 (4.4)
Average over first 24 h	0.4 (0–1)	3 (4.4)
At discharge	0 (0–1)	6 (8.8)
Opioid consumption		
Median MME (IQR) before SPING block	15 (4–30)	4 (5.9)
Median MME (IQR) after SPING block	0 (0–0)	1 (1.5)
Mobility Outcomes		
Able to sit upright after SPING block		1 (1.5)
Yes	57 (83.8)	
No	10 (14.7)	
Mobility at discharge		0
Bedridden	21 (30.9)	
Able to sit upright	15 (22.1)	
Able to perform bed–chair transfers	30 (44.1)	
Able to take a couple of steps	2 (2.9)	
Safety Outcomes after SPING block		
SPING block effectiveness	63 (92.6)	0
Repeat procedures	5 (7.4)	0
Complications during hospital admission after SPING block		0
Hypotension	4 (5.9)	
Fecal incontinence	3 (4.4)	
Urinary incontinence	2 (2.9)	
Fever	2 (2.9)	
Cardiac decompensation 1 day after SPING block	1 (1.5)	
Numbness and continued cries from patient	1 (1.5)	
Pain with turning movement	1 (1.5)	
Paresis other leg	0	
Complications at follow-up after SPING block		0
Possible neuropathic pain or decreased efficacy	1 (1.5)	
Pain knee	1 (1.5)	
Neuropathic pain (after 56 days)	1 (1.5)	
Slight unilateral groin pain	1 (1.5)	
Process Outcomes		
Median time (IQR) from ED admission until SPING block, days	1 (0–2)	0
Median time (IQR) from SPING block until discharge, days	1 (0–3)	4 (5.9)
Discharge location after SPING block		0
Stayed in hospital	5 (7.4)	
Community dwelling, independent	1 (1.5)	
Community dwelling, with ADL care	5 (7.4)	
Nursing home	47 (69.1)	
Hospice	6 (8.8)	
Geriatric rehabilitation	4 (5.9)	
Long-Term Patient Outcomes		
Survival		
Median survival (IQR) after SPING block, d	13 (7–44)	19 (27.9)
30-day mortality after SPING block	32 (47.1)	19 (27.9)
90-day mortality after SPING block	40 (58.8)	19 (27.9)
Follow-up		
Median follow-up duration (IQR), d	146 (40–372)	6 (8.8)
Residual pain		3 (4.4)
Yes	15 (22.1)	
No	50 (73.5)	
Satisfied with SPING block		3 (4.4)
Yes	55 (80.9)	
No	10 (14.7)	

Note: Data are given as numbers (percentage), unless otherwise indicated. Abbreviations: NRS, Numeric Rating Scale; IQR, Inter Quartile Range (denoted as p25–p75); SPING, Spinal Phenol IN Glycerol; MME, Morphine Milligram Equivalents; ADL, activities of daily living; ED, emergency department.

## Data Availability

The data presented in this study are available on request from the corresponding author. The data are not publicly available because of ethical restrictions.

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
