# Peer review of "SPING Block Analgesia in Non-Operative Management of Proximal Femur Fractures in Older Adults Living with Frailty: A Retrospective Cohort Study"

_geriatrics, 2025, doi:10.3390/geriatrics10010010_

Round 1

Reviewer 1 Report

Comments and Suggestions for Authors

The Authors aimed safety and outcomes of Spinal Phenol IN Glycerol block in frail older adults with proximal femur fractures (PFF) managed non-operatively. 

The topic is interesting. Howver, the study presents makpr biases.

Follow up extremely short. Which was mortaliity rate? The series is heterogenous in terms of fracture pattern. Inclusion of periprosthetic fractures: please define which type. How many patients with periprosthetic/subtrochanteric fractures required blood transfusions?

Please define indications to non-operative treatment. This might include major biases.

Table 1 not clear

How was SPING managed over time at home??Till death? Did any patient (with medial fracture) returned to deambulate? How many were able to sit?

Conclusions are not supported by results.

Author Response

Reviewer 1:

Comment:
The Authors aimed safety and outcomes of Spinal Phenol IN Glycerol block in frail older adults with proximal femur fractures (PFF) managed non-operatively.
The topic is interesting. Howver, the study presents makpr biases.

RESPONSE: Thank you for taking the time to review our manuscript and for recognizing the relevance and interest of this topic. We fully acknowledge that biases may arise (and indeed are inherent) in this type of research. Below, we address the specific comments regarding bias in further detail.

Comment: Follow up extremely short. Which was mortaliity rate?
RESPONSE: We acknowledge that the follow-up period in this study was relatively short, which may limit the assessment of long-term outcomes. However, since the decision to perform SPING is part of a multidisciplinary shared decision-making process, it is generally applied to patients who are most likely to benefit from non-operative management (as defined in the FRAIL HIP study). For this reason, mortality was not initially considered the primary outcome of interest.

That said, we understand and agree with the reviewer that mortality rates are of significant interest to readers. It is important to note, however, that these rates primarily reflect the outcomes of the shared decision-making process rather than the direct effects of SPING itself.

We assessed mortality in the follow-up, which is described in section ‘2.5 Follow-up’. We asked patient representatives about the date of death, to enable mortality assessment despite the short hospital admission duration. We have emphasized this in the manuscript following your feedback. Mortality rates are described in ‘table 3 Patient outcomes’.

COMMENT: The series is heterogenous in terms of fracture pattern. Inclusion of periprosthetic fractures: please define which type.

RESPONSE: We thank the reviewer for highlighting this point, as the heterogeneity of our series was indeed intentional. We consider this a strength of our study, as SPING appears to be applicable to all types of hip fractures. This distinguishes it from approaches like PENG, which may be less effective for fractures located further from the hip capsule.  That said, the reviewer makes a valid point that fracture type is an important factor to consider in future studies. We have already addressed this to some extent in the discussion section of the manuscript, referencing the study by Verduijn et al. in JAMDA 2024, which emphasizes the significance of fracture type. We appreciate the opportunity to expand on this point and acknowledge its relevance for further research.

CHANGES MADE TO MANUSCRIPT: We have added the types of periprosthetic fractures to the patient characteristics in ‘Table 1 Patient Characteristics’ and included the heterogeneity of the study population in the limitations section. We have also emphasized the applicability of SPING block across the various fracture types in the manuscript.

COMMENT: How many patients with periprosthetic/subtrochanteric fractures required blood transfusions?

RESPONSE: Due to the retrospective character of the study, this information was not available. Anemia or bleeding complications were however not reported as complications of this treatment. Furthermore, the hip fractures are relatively low level trauma, e.g. falling out of bed, and thus are less likely to require blood transfusions in comparison to, for example, high level trauma such as traffic accidents.

CHANGES MADE TO MANUSCRIPT: We included this in the manuscript.

COMMENT: Please define indications to non-operative treatment. This might include major biases.

RESPONSE: We acknowledge the concern regarding potential biases associated with the selection of patients for non-operative treatment. As mentioned in our response to the mortality rate comment, such selection is inherent to the decision-making process, which is multidisciplinary and based on shared decision-making (SDM). It is indeed likely that “healthier” patients are more often selected for surgery, while non-operative management is reserved for patients deemed less suitable for surgical intervention. 
That being said, we prefer not to frame this as “bias” but rather as patient selection. To provide consistency and transparency, we applied the FRAIL-HIP study criteria to define indications for non-operative treatment. Additionally, we have thoroughly described patient characteristics in the results section and Table 1 to give readers a clear understanding of the study population. For example, Table 1 highlights the frailty of this cohort, including a high prevalence of cognitive impairments (8.8%), dementia (58.8%), nursing home residency (>70%), and ASA 4 classification. 
Optimizing this selection process is an important aim of future research. In fact, as part of a recently awarded grant, we are planning studies to investigate how this process can be refined and standardized. Additionally, we aim to explore the extent to which patient factors influence the success (or lack thereof) of SPING. We believe this is a crucial step in ensuring that non-operative management is applied effectively and consistently in the future. 

We hope this explanation adequately addresses the reviewer’s concern and provides important context regarding the study’s methodology and population.

CHANGES MADE TO MANUSCRIPT:We have pointed  out the use of the FRAIL HIP criteria in the section ‘2.2 Procedures’.

Comment: Table 1 not clear

RESPONSE: Thank you for pointing this out. Table 1 is part of the methods section and outlines how the various measurements were conducted. Instead of presenting this information in textual form, we have chosen to organize it in a table for clarity and ease of reference.

CHANGES MADE TO MANUSCRIPT: We have shortened the table to point out the most relevant variables. The full table has been made available in the Appendix.

COMMENT: How was SPING managed over time at home??Till death? Did any patient (with medial fracture) returned to deambulate? How many were able to sit?

RESPONSE: Thank you for raising these  questions, which underscore the need to provide additional clarity for readers. First, we would like to emphasize that SPING is a one-time, definitive intervention. As such, no ongoing management is required after the procedure. This represents one of the key advantages of SPING compared to other nerve blocks or catheter-based approaches that require continuous pain medication. These alternatives often face challenges such as catheter dislocation or obstruction, which can necessitate the return of a highly frail patient population -often from their home or nursing home- to the hospital for further intervention. Mobility outcomes are reported in Table 3.

CHANGES MADE TO MANUSCRIPT: We have emphasized the added value of SPING block as an one-time and definitive intervention in both the introduction as the discussion.

COMMENT: Conclusions are not supported by results.
RESPONSE: Thank you for your comment. While we are not entirely certain about the specific concern raised, we have adjusted and softened the conclusions to ensure they are more closely aligned with the results presented. That said, we maintain our position that SPING has the potential to provide significant added value. To further substantiate this, we are planning broader studies supported by a recently awarded grant, as mentioned in our response to other comments. These future studies will allow us to investigate SPING's effectiveness and impact in greater depth.

CHANGES MADE TO MANUSCRIPT: We have made adjustments to the wording of our discussion, to align it better with the methods and results of our study.

Reviewer 2 Report

Comments and Suggestions for Authors

Dear Authors, 

I was pleased to review the paper entitled " SPING Block Analgesia as Non-Operative Management of Proximal Femur Fractures in Frail Older Adults: A Retrospective Cohort Study" - 

- MDPI –

The present paper is very interesting, it focuses on a relevant clinical scenario, for orthopedics, potentially influencing the surgical and clinical practice for the management of Proximal Femur Fractures. 

Therefore, it is my opinion that the content is original, current, and relevant. 

Thus, there are some minor remarks:

- Title: The title gives a fine idea of the topic to be covered.

- Abstract: correct.

- Introduction: While the introduction sets the stage well, it could benefit from explicitly stating the research gap this study addresses, especially regarding the lack of robust data for SPING blocks in palliative care.

Better describe fracture of the proximal femur, a fracture that frequently occurs due to low-energy trauma such as falls. Surgical interventions, including anterograde nailing or dynamic hip screws (DHS), are critical in the management of these fractures (you could add from doi: 10.1302/2633-1462.56.BJO-2023-0163.R1)

Clearly state the hypothesis or primary research question.

- Method: add references of fracture type as in table 1.

The use of the table is very helpful to the reader. i request that only the endpoints of the study and your hypothesis be written in the final part of the methods. 

The study did not include a control group, explain the reason why.

- Results: add “before SPING block” to table 3.

- Discussion: describe other alternative techniques to the one studied that could be the subject of future publication.

Describe the strengths of your research and future implications.

The paper generally is well written and needs only minor changes.

Author Response

Reviewer 2:

COMMENT: I was pleased to review the paper entitled " SPING Block Analgesia as Non-Operative Management of Proximal Femur Fractures in Frail Older Adults: A Retrospective Cohort Study" -
The present paper is very interesting, it focuses on a relevant clinical scenario, for orthopedics, potentially influencing the surgical and clinical practice for the management of Proximal Femur Fractures.
Therefore, it is my opinion that the content is original, current, and relevant. Thus, there are some minor remarks:

RESPONSE: Thank you for taking the time to review our manuscript and for your valuable comments.

COMMENT:
- Title: The title gives a fine idea of the topic to be covered.
- Abstract: correct.

RESPONSE: Thank you.

COMMENT:
- Introduction: While the introduction sets the stage well, it could benefit from explicitly stating the research gap this study addresses, especially regarding the lack of robust data for SPING blocks in palliative care.

RESPONSE: Thank you for highlighting this point. We agree that explicitly stating the research gap would strengthen the introduction.
CHANGES MADE TO MANUSCRIPT: We have revised the introduction to clearly address the lack of robust data regarding SPING blocks in palliative care.

COMMENT:
Better describe fracture of the proximal femur, a fracture that frequently occurs due to low-energy trauma such as falls. Surgical interventions, including anterograde nailing or dynamic hip screws (DHS), are critical in the management of these fractures (you could add from doi: 10.1302/2633-1462.56.BJO-2023-0163.R1)

RESPONSE: Thank you for bringing this to our attention. While we agree that this is important in surgical management, the focus of our study lies with the non surgical, non-operative management (NOM) of hip fractures. Therefore, we do not elaborate on surgical interventions as it is not within our aim.

CHANGES MADE TO MANUSCRIPT: None.

COMMENT: Clearly state the hypothesis or primary research question.
RESPONSE: Thank you for the suggestion.
CHANGES MADE TO MANUSCRIPT: We have now explicitly stated our primary research question in the last paragraph of the introduction.

COMMENT: Method: add references of fracture type as in table 1.

RESPONSE: Thank you for pointing this out.
CHANGES MADE TO MANUSCRIPT: We have now added the reference on fracture type in the methods section ‘2.1 Study design and setting’.

COMMENT: The use of the table is very helpful to the reader. i request that only the endpoints of the study and your hypothesis be written in the final part of the methods.
RESPONSE: Thank you for pointing out that this need for clarification.
CHANGES TO MANUSCRIPT: We have now elaborated on outcome measures in the methods section ‘2.4 Data collection and outcomes’.

COMMENT: The study did not include a control group, explain the reason why.

RESPONSE: Thank you for pointing this out. The study did not include a control group because it was designed as a retrospective cohort study focusing on the feasibility and outcomes of SPING blocks in a specific, highly vulnerable/frail population. SPING block for patients opting for NOM standard of care in our hospital. Given the ethical and practical challenges of randomizing frail older adults with proximal femur fractures, particularly those managed with a palliative intent, we opted for an observational approach to evaluate real-world application and effectiveness. Additionally, the nature of the shared decision-making process inherent in these cases precludes the standardization required for a controlled comparison. Future research, supported by our recently awarded grant, will aim to address these limitations by exploring more comprehensive methodologies to better compare outcomes.

CHANGES TO MANUSCRIPT: We have now elaborated on this decision in the methods section ‘2.1 Study design and setting’. We have also included the lack of control group in the observational design as a study limitation in the discussion section.

COMMENT: Results: add “before SPING block” to table 3.

RESPONSE: Thank you for pointing out this need for clarification.
CHANGES TO MANUSCRIPT: We altered Table 3 and  have elaborated on the time measurements were taken, that is before or after SPING block.

COMMENT: Discussion: describe other alternative techniques to the one studied that could be the subject of future publication.

RESPONSE: We described the techniques, known to the authors, that have been described for application in a palliative setting for patients with hip fractures.
CHANGES TO MANUSCRIPT: We have pointed out that PENG block and PHPN may be examples of the potential of regional nerve blocks in the palliative setting for patients with hip fractures.

COMMENT: Describe the strengths of your research and future implications.

RESPONSE: The strengths of our study are the uniform application or SPING block, the sample size, and the averaging of pain scores. It adds to current literature by describing this intervention in a respectable cohort of patients. Additionally, the results of this study are likely to be generalizable, as we included the specific target group of this treatment in our study.
CHANGES TO MANUSCRIPT: We have added the generalizability of this study in this specific population. Due to the observational design, we assessed SPING block in the target population, which increases the generalizability of our results in this specific population, which is expected to be similar across hospitals.

COMMENT: The paper generally is well written and needs only minor changes.

RESPONSE: We thank the reviewer once again for this positive evaluation.

Reviewer 3 Report

Comments and Suggestions for Authors

Introduction/overall

add context on SPING blocks theoretical advantages over PENG / other blocks

Highlight the role of SDM in aligning SPING block and palliative goals

add some details on how frailty impacts the choice of nonoperative treatments

Methods

this section should include more robust methods/ explanation of the method to assess pain in patients with severe cognitive impairment

Address the absence of a control/comparison group(PENG or other nerve blocks)

add details on how ethical considerations influenced the design

Results

Authors should describe the potential for SPING block to improve QOL despite high mortality

how opioid reduction contributes to decreased delirium and improved alertness?

What are the implications of early discharge on posthospital pain and complications?

Relate satisfaction rates to the palliative goals of the treatment in the results section

Author Response

Reviewer 3:

COMMENT: add context on SPING blocks theoretical advantages over PENG / other blocks

RESPONSE: Thank you for reviewing our manuscript and providing valuable feedback. We believe SPING offers several theoretical advantages over PENG and other blocks. First, SPING is a one-time, definitive intervention. Second, unlike PENG, which logically depends on its proximity to the hip capsule for effectiveness, SPING is not constrained by this anatomical relationship. Third, SPING does not rely on repetitive interventions or the maintenance of devices such as catheters, which can dislocate or malfunction.
CHANGES TO MANUSCRIPT: We have emphasized the one-time, definitive character of SPING block in the introduction.

COMMENT: Highlight the role of SDM in aligning SPING block and palliative goals

RESPONSE: Thank you for this insightful comment. We agree that SDM plays a crucial role in aligning the use of SPING blocks with the goals of palliative care. SDM ensures that the intervention aligns with the patient's preferences, clinical context, and overarching palliative care objectives. This process is particularly important for tailoring treatment in frail older adults and ensuring that SPING blocks are applied in a way that optimally supports comfort and quality of life.
CHANGES TO MANUSCRIPT: We have now emphasized this in the section ‘2.2 Procedures’ and in the discussion section.

COMMENT: add some details on how frailty impacts the choice of nonoperative treatments

RESPONSE: Thank you for pointing this out. Frailty is a major contributor in the decision to refrain from surgery.
CHANGES TO MANUSCRIPT: We have elaborated on patient selection in section ‘2.2 Procedures’, we have also included the issue on patient selection in the discussion section.

COMMENT:
Methods
this section should include more robust methods/ explanation of the method to assess pain in patients with severe cognitive impairment

RESPONSE: The method of pain assessment is indeed an important consideration in the study design. Due to the retrospective design, we were not able to use pain measurements recommended for patients living with cognitive impairment or dementia. It was necessary to include these patients, because our research question regarded the effect of SPING block in the older patients living with frailty. This is a limitation of our study, as described in the discussion section. We recommend the use of fitting pain assessment methods, such as PACSLAC, for future prospective studies.

CHANGES TO MANUSCRIPT: We have elaborated in ‘Table 1 Outcome measures, operationalization, measurement instruments and timepoints’ that only available NRS scores could be recorded.

COMMENT: Address the absence of a control/comparison group (PENG or other nerve blocks)

RESPONSE:  As SPING block is standard of care in the study hospital, this observational study did not allow for a control group not receiving SPING. We have addressed this in our methods and discussion section.
CHANGES TO MANUSCRIPT: We elaborated this in the method section ‘2.1 Study design and setting’.

COMMENT: add details on how ethical considerations influenced the design

RESPONSE: As previously mentioned, SPING block is standard of care in the study hospital. We did not deem it ethical to deny patients local pain management for study purposes.

CHANGES TO MANUSCRIPT: We elaborated this in the method section ‘2.1 Study design and setting’.

COMMENT: results

Authors should describe the potential for SPING block to improve QOL despite high mortality

RESPONSE: Thank you for pointing this out. The potential of SPING block to improve quality of life is indeed an important consideration.
CHANGES TO MANUSCRIPT: We have highlighted this in the discussion section, as elaborated also further in the next comment.

COMMENT: how opioid reduction contributes to decreased delirium and improved alertness?

RESPONSE: As we describe in our introduction, a known side effect from opioid use includes the risk of delirium and decreased consciousness in older adults living with frailty. This was an important rationale for developing the SPING block. In this study design, we hypothesized that pain reduction through SPING block would result in decreased opioid need with subsequent side effects. The retrospective design with short follow-up did not allow for reliable and valid assessment of delirium rates or level of consciousness. Based on this study we cannot state that SPING block decreases delirium or improves alertness. However, it is likely that reduction of opioid use results in less delirium and increased consciousness. Further research is necessary to validate this hypothesis.

CHANGES TO MANUSCRIPT: We have described this in the discussion section.

COMMENT: What are the implications of early discharge on posthospital pain and complications?

RESPONSE: Early discharge is part of our standard hospital protocol, which aims to enable patients to reside in a trusted and comfortable environment in their last days. However, hospital discharge was only granted when it was safe and pain relief was sufficient. Early discharge should always be accompanied with careful consideration before discharge and close follow-up assessment, as we describe in the discussion section. For this study, early discharge confronted us with a gap in the available data on post-hospital pain and possible complications. We anticipated on this by telephone follow-up after discharge. Future research has to evaluate this concern, as it may not be ethical to refrain a patient from their preferred residency in their last days, to collect study data. Future research should aim for a hybrid solution with close collaboration between patient carers and the researchers.

CHANGES TO MANUSCRIPT: We have clarified the recommendations for early discharge and the recommendations for future research.

COMMENT: Relate satisfaction rates to the palliative goals of the treatment in the results section

RESPONSE: Satisfaction was attributed to the SPING block treatment, not to palliative care goals. This was previously unclear in the discussion section. Relating satisfaction rates was not a goal of this study.
CHANGES TO MANUSCRIPT: We have clarified our interpretation of the high satisfaction rates in the discussion section. 

Round 2

Reviewer 1 Report

Comments and Suggestions for Authors

Unfortunately, despite Authors'efforts, they were not able do address appropriately to most of my previous concerns.

In particular, I regret they decided to maintain such different fracture types. Periprosthetic hip fractures must be removed from analysis, as they should be considered separately. The series is extremely heterogeous and brings major biases.

Comments on the Quality of English Language

Still many grammar and syntax errors.

Author Response

We thank the reviewer for their re-evaluation of our manuscript. As explained in our first rebuttal, we respectfully disagree and consider the inclusion of the different fracture types a strength of this initial series. Of course, future studies (which we plan to conduct using the grant we have secured) will need to determine whether patients with various fracture types are indeed suitable candidates for SPING.

Reviewer 3 Report

Comments and Suggestions for Authors

Congratulations! The authors have answered all my querries and the work can now be published.

Author Response

We thank the reviewer for their re-evaluation of our manuscript.